# Fish Oil Supplementation Mitigates High-Fat Diet-Induced Obesity: Exploring Epigenetic Modulation and Genes Associated with Adipose Tissue Dysfunction in Mice

**DOI:** 10.3390/ph17070861

**Published:** 2024-07-01

**Authors:** Jussara de Jesus Simão, Andressa França de Sousa Bispo, Victor Tadeu Gonçalves Plata, Lucia Maria Armelin-Correa, Maria Isabel Cardoso Alonso-Vale

**Affiliations:** 1Post-Graduate Program in Chemical Biology, Institute of Environmental Sciences, Chemical and Pharmaceutical, Federal University of São Paulo—UNIFESP, Diadema 09913-030, Brazil; jussara.simao@unifesp.br (J.d.J.S.); andressa.franca@unifesp.br (A.F.d.S.B.); victor.plata@unifesp.br (V.T.G.P.); lucia.correa@unifesp.br (L.M.A.-C.); 2Department of Biological Sciences, Institute of Environmental Sciences, Chemical and Pharmaceutical, Federal University of São Paulo—UNIFESP, Diadema 09913-030, Brazil

**Keywords:** obesity, inflammation, H3K27, n-3 PUFA, WAT, leptin

## Abstract

This study investigated the effects of fish oil (FO) treatment, particularly enriched with eicosapentaenoic acid (EPA), on obesity induced by a high-fat diet (HFD) in mice. The investigation focused on elucidating the impact of FO on epigenetic modifications in white adipose tissue (WAT) and the involvement of adipose-derived stem cells (ASCs). C57BL/6j mice were divided into two groups: control diet and HFD for 16 weeks. In the last 8 weeks, the HFD group was subdivided into HFD and HFD + FO (treated with FO). WAT was removed for RNA and protein extraction, while ASCs were isolated, cultured, and treated with leptin. All samples were analyzed using functional genomics tools, including PCR-array, RT-PCR, and Western Blot assays. Mice receiving an HFD displayed increased body mass, fat accumulation, and altered gene expression associated with WAT inflammation and dysfunction. FO supplementation attenuated these effects, a potential protective role against HFD-induced obesity. Analysis of H3K27 revealed HFD-induced changes in histone, which were partially reversed by FO treatment. This study further explored leptin signaling in ASCs, suggesting a potential mechanism for ASC dysfunction in the obesity-rich leptin environment of WAT. Overall, FO supplementation demonstrated efficacy in mitigating HFD-induced obesity, influencing epigenetic and molecular pathways, and shedding light on the role of ASCs and leptin signaling in WAT dysfunction associated with obesity.

## 1. Introduction

The high-fat diet (HFD) obesity induction model triggers inflammation and promotes metabolic dysfunction, contributing to the development of obesity-related complications such as insulin resistance and cardiovascular diseases [1,2,3]. One notable characteristic, particularly in the context of white adipose tissue (WAT), is the presence of a chronic state of low-grade inflammation, often termed “metaflammation”, which plays a key role in obesity-related health issues [3,4,5,6,7]. Understanding how inflammation is modulated during obesity can provide valuable insights into the underlying mechanisms of these conditions and identify or guide the development of therapies targeted at specific inflammatory pathways exacerbated by obesity, leading to new approaches for its treatment and management.

Natural and synthetic treatments are gaining popularity in managing high-fat diet (HFD)-induced obesity, such as those increasing insulin sensitivity and secretion while reducing glucose levels [8,9]. Along the same line, the beneficial effects of long-chain polyunsaturated fatty acids n-3 (n-3 PUFA), especially eicosapentaenoic acid (EPA) and docosahexaenoic acid (DHA) found in fish oil (FO), have been widely demonstrated in improving dyslipidemia, glucose tolerance, insulin sensitivity, and reducing adipose mass in HFD-induced obesity models [10,11,12,13].

Our previous studies demonstrated that obese animals supplemented with FO (8 weeks) containing EPA in a 5:1 ratio consistently showed reductions in body weight and adipose mass, along with improvements in adipocyte function and reductions in metabolic and endocrine dysfunctions associated with obesity. FO supplementation also decreased the expression of pro-inflammatory cytokines, suggesting its potential in mitigating obesity-related endocrine disorders [14,15,16,17], as well as a role in epigenetically regulating the inflammation pathway in WAT cells.

It is well known that the proper functioning of WAT requires continuous remodeling to accommodate its rapid and dynamic ability to expand or contract, adjusting lipid stores. This process involves adipocyte differentiation and WAT plasticity, which ensures the supply of nutrients and oxygen to WAT, as well as appropriate production of adipokines [18,19]. Adipose tissue-derived mesenchymal stem cells (ASCs) are abundant in WAT, undergoing adipogenesis via epigenetic changes, particularly involving histone 3 lysine 27 (H3K27) marks deposition [19,20,21,22]. Trimethylation of H3K27 (H3K27me3) by EZH2 silences genes (16) and has been shown to promote adipogenesis by up-regulating PPARγ and repressing Runx2, with these effects being reversed by the H3K27me3 demethylases KDM6A and KDM6B [21,23,24]. Conversely, acetylation (H3K27ac) by CREBBP and EP300 (Cbp/p300) activates transcription and has been reported to promote adipogenesis through interaction with PPARγ [25]. The proper action of H3K27 modifiers in promoting adipogenesis not only facilitates the formation of healthy adipose tissue but also plays a protective role against the adverse effects of hypertrophic obesity, helping to maintain metabolic balance and reducing the risk of chronic inflammation and its associated complications [21,26]. However, the impact of H3K27 marks on WAT in the context of chronic inflammatory states during obesity remains poorly understood. Recent studies have shed light on the role of H3K27 in regulating inflammatory responses in fibroblasts and monocytic cells [27,28]. For instance, IFN-γ and LPS were shown to synergistically induce CCL2 expression in monocytic cells via H3K27 acetylation [27]. Another study demonstrated that repressive H3K27 trimethylation is associated with hyperinflammation in fibroblasts [28]. These findings suggest potential novel targets for therapeutic intervention.

In any case, the dynamic nature of histone modifications, such as H3K27ac and H3K27me3, plays a pivotal role in regulating gene expression in response to various environmental and physiological cues. Histone acetylation, in particular, stands out as an exceptionally dynamic chromatin modification. It is greatly influenced by the availability of acetyl-CoA [29,30,31], with the addition of acetyl groups to lysine residues (such as H3K27ac) resulting in a more open chromatin structure conducive to gene transcription [32]. The ATP citrate lyase (ACL) enzyme catalyzes the conversion of citrate and coenzyme A (CoA) into oxaloacetate and acetyl-CoA. Consequently, this enzyme plays a crucial role in generating acetyl-CoA, a pivotal precursor in numerous metabolic pathways, in addition to serving as a substrate for histone acetyltransferases [33], thereby connecting metabolism to protein acetylation processes. Carrer and colleagues demonstrated that a 4-week consumption of an HFD, known to suppress ACL [34,35,36], resulted in decreased acetyl-CoA levels in the whole WAT [34]. Moreover, ACL has been shown to regulate histone acetylation levels and influence gene regulation [31,37,38]. The regulation of ACL expression in ASCs remains to be explored, particularly in the context of obesity. Interestingly, leptin signaling has been shown to reduce ACL expression in hepatocytes and adipocytes [35]. However, whether leptin (which is abundantly expressed in WAT of obese animals) acts paracrinally on ASCs to influence ACL expression, which in turn, could epigenetically mediate differentiation and/or inflammation in these cells to influence dysfunction of WAT, remains to be explored.

Few studies have explored the correlation between the elevated products of chronic low-grade inflammation in individuals with obesity and the expression of the H3K27 modifiers and/or ACL. Similarly, there is a lack of research on the impact of HFD, with or without FO supplementation, on the levels of H3K27ac and H3K27me3 in WAT. In the present study, the inflammation pathway mediated by chemokine and cytokine signaling emerged as the most prominently affected signaling pathway in our HFD-induced obesity murine model. Significantly, this pathway was modulated by FO treatment. We sought to investigate whether this modulation involves epigenetic modifications of histone 3 lysine 27 (H3K27), as well as the participation of leptin and ACL in the process. To address this question, we employed a combination of functional genomics tools, including RT-PCR, PCR-array, along with Western Blot analyzes in a murine obesity model.

## 2. Results

### 2.1. Obesity Model Characterization

Obesity parameters were assessed at the end of the 12-week experimental protocol for obesity induction. As expected, there was a decrease in food and caloric intake but an increase in fat intake (~5×, Figure 1A) in mice receiving the HFD. Regarding body mass, a statistical difference between the control group and the groups of animals receiving the HFD became evident from the 3rd week. This difference increased throughout the weeks of the experimental protocol, and by the 12th week of HFD administration, the animals receiving it showed a ~3.5× body mass gain compared to those receiving the CO diet (Figure 1B).

Figure 1C–E also illustrates values related to fasting blood glucose and the glucose tolerance test (GTT). Compared to the CO diet, mice fed with the HFD showed an approximately 50% increase in fasting blood glucose (Figure 1C) and glucose intolerance (Figure 1D), with an expressive (82%) increase in the area under the curve (Figure 1E). Thus, our diet-induced obesity model was characterized, confirming our previous results [15,17,39]. We subsequently supplemented the animals with FO (HFD + FO group) in the last 8 weeks of the 16-week experimental protocol with the HFD.

### 2.2. WAT depots Extracted from Mice Treated with CO, HFD, and HFD + FO Diets: Depot Mass and Gene Expression by PCR Array

As expected [15,17], mice consuming the HFD presented a significant increase of 50% in body mass when compared to the CO group, and the HFD + FO group was still 40% higher in relation to the CO group, but presented a significant reduction of ~30% (*p* < 0.05) compared to the HFD group (Figure 2A). Moreover, animals in the HFD group exhibited a significant increase in the mass of the visceral epididymal (Epi) fat depot (by ~3×), while treatment with FO completely prevented this increase in fat mass (Figure 2B). However, no significant difference was observed between the HF and HFD + FO groups in the mass of the visceral retroperitoneal (Rp) (Figure 2C) or subcutaneous inguinal (Ing) (Figure 2D) fat depots. The complete metabolic characterization of FO treatment in this model, along with measurements of other tissue weights such as the liver and interscapular brown adipose tissue, has been previously demonstrated and reported in studies conducted by our group [14,15,16,17].

An analysis of Epi WAT depots (whole tissues) was conducted to explore their gene expression profiles. To do so, we designed a customized panel of 84 essential genes for adipose tissue using a PCR array gene expression assay. Following the validation of reactions via array control plates, an in-depth comparative analysis was performed on the tissue Ct values to thoroughly examine the gene expression changes induced by the HFD, associated (or not) with FO supplementation. Initially, our emphasis was on scrutinizing genes with differential expression, both up- or down-regulated, attributed to the HFD when compared to the CO diet. These findings, contrasting obese animals with controls, are comprehensively outlined in Table 1.

Among the various genes modulated, we highlight the up-regulation of the genes *Lep* (+25.48×), *Ncor2* (+3.08×), *Ccl2* (+5.07×), *Tnf* (+10.65×), *Nfkb1* (+2.52×), and *Cd68* (9.16×), triggered by the HFD when compared to the CO group (Table 1). With the exception of *Ncor2* (anti-adipogenic factor), all these genes encode cytokines (*leptin*, *Tnfa*) and macrophage chemotactic factors (*Ccl2/Mcp-1*) or markers of macrophages (*Cd68*) and inflammation pathways (NFkB). There was also an increase in *Dio2* (+3.98×) and *Elovl3* (+2.63×), indicating an increase in diet-induced thermogenesis. 

We also observed that the HFD suppressed the expression of numerous genes, including those encoding adipokines (adiponectin and resistin), enzymes involved in lipid biosynthesis (*Acc1*, *Scd-1*, *Lipin1*, *Pck1*, and *Fas*), factors related to adipogenesis (*Cebpa*, *Cebpd*, *Fabp4/Ap2*, *Fgf-2*, *Fgf-10*, *AP-1/C-Jun*, *Sfrp1*, *Klf15*, *Adrb2*, *Dlk1/Pref-1*, *Foxo-1*, *Shh*, *Wnt1*, *Wnt3a*, *Gata2*), proteins associated with browning, thermogenesis, and fatty acid oxidation (*Bmp7*, *Pgc1-a*, *Pgc1-b*, *Sirt3*, *Tbx1*, *Ucp1*, *Nr1h3*, *Wnt10b*), adipocyte receptors (*Lepr*, *Adipor2*, *Adrb1*), pro- and anti-inflammatory cytokines (*Ifn-γ*, *Il1-β*, *Il-6*, *Il-13*), and insulin signaling pathway proteins (*Insr*, *Irs-1*, *Irs-2*, *Pik3r1*). 

The data concerning both up-regulated and down-regulated genes in the samples from the HFD and HFD + FO groups are visualized in a heatmap (clustergram, Figure 3). It was evident that the treatment of obese mice with FO altered the extensive list of genes negatively affected by the HFD, inducing an increase in the expression of most of those genes (and others). These encompass genes encoding adipokines (adiponectin, leptin, and resistin), lipolytic and lipogenic enzymes (*Scd-1*, *Lpl*, *Lipin1*, *Pck1*), pro-adipogenic factors (*Cebpa*, *Pparg*, *Srebp-1*, *Fabp4/aP2*, *Sfrp1*, *Glut4*), proteins involved in browning and thermogenesis, mitochondrial biogenesis, and fatty acid oxidation (Dio-2, Elovl3, Foxc2, Ppar-α, Pgc1-α, Tbx1, Tfam, Ucp-1), pro- and anti-inflammatory cytokines (Mcp-1, Cxcl10, Il1-Β, Il-4, Il-6, Il-13, Resistin), and signaling transduction factors (Irs-1, Nf-kb). 

### 2.3. Expression of H3K27 Modifiers, H3k27ac, H3k27met3, and Acly/ACL in the Visceral Epi WAT from Mice

Regarding possible alterations in epigenetic marks, we initially assessed the expression of genes encoding enzymes responsible for acetylation (*Ep300* and *Crebbp*), as well as methylation (*Ezh2*) and demethylation (*Kdm6a* and *Kdm6b*) of H3K27 in visceral Epi WAT of control mice, obese mice (HFD), and obese mice treated with FO (HFD + FO). There was no statistically significant difference observed in the transcript levels of *Ezh2* (Figure 4B). However, the genes encoding the acetylases *Crebbp* (Figure 4C) and *Ep300* (Figure 4D), as well as the demethylase *Kdm6b* (Figure 4F), demonstrated a notable increase in expression within the HFD group compared to the control. This effect was completely reversed by FO treatment for *Crebbp* and *Ep300*, while partially reversed for *Kdm6b*. Additionally, no difference was noted between the groups in the expression of the gene encoding the enzyme *Kdm6a* (Figure 4E).

We next investigated the expression of the *Acly* (gene) and ACL (protein), as well as histone-modifying proteins. Consistent with the decreased *Acly* expression (Figure 4A), we observed a reduction in ACL expression in the visceral Epi WAT of obese animals (Figure 5A). Corroborating these findings, we detected a decrease in the expression of H3K27ac in the group of animals with obesity induced by the HFD (Figure 5B), and once again, FO was able to completely prevent this effect. Finally, a significant decrease in the expression of the H3K27me3 protein was also observed (Figure 5C), an effect partially prevented by FO.

### 2.4. Gene Expression of Acly and Leptin Receptors in ASCs from Mice

In Figure 6, we validated the expression of *Lepr1* and *Lepr2* receptor isoforms in ASCs. We observed that ASCs extracted from the visceral WAT showed no difference in receptor isoform expression between the CO and HFD animal groups (Figure 6A,B). Moreover, *Lepr3* isoform was not expressed in these cells. When we treated these mice ASCs with leptin (in vitro for 24 h) and examined the expression of LEP protein receptor (long isoform, Ob-Rb, responsible for total signal transduction), we found that ASCs fully expressed these receptors, whose expression remained unchanged when the ASCs were exposed to leptin (Figure 6C). Finally, we evaluated the expression of the *Acly* gene in these ASCs treated in vitro with leptin. Interestingly, we observed a reduction in Acly expression in the HFD mice group. This finding suggests that, in a leptin-rich medium, ASCs exhibit a decrease in *Acly* expression (Figure 6D).

## 3. Discussion

We investigated whether FO treatment, rich in EPA, protects against the inflammation pathway triggered by HFD-induced obesity in murine WAT. Activation of this pathway is associated with changes in the expression of several important genes involved in WAT metabolism and cell differentiation. Additionally, we explored whether these effects are modulated by H3K27 modifications and the impact of leptin (whose secretion is high in the WAT of obesity individuals) on ASCs. Our results suggest that FO mitigates the negative effects of chronic inflammation associated with obesity, and that its effects involve epigenetic mechanisms by modulating ACL expression and H3K27 acetylation. This study also underscores the role of leptin not only as an important endocrine signal but also as a paracrine one, leading to significant impact on ASCs, which ultimately could affect their adipogenic potential.

Mice fed the HFD for 16 weeks displayed a substantial increase in the mass of visceral fat depots and an up-regulation of genes encoding cytokines, macrophage chemotactic factors, markers of macrophages, and inflammation pathways, among others. These findings align with the existing literature describing numerous pronounced and detrimental effects of the HFD [1]. We also observed a down-regulation in the expression of numerous genes, including those encoding adiponectin, enzymes involved in lipid biosynthesis, factors and proteins related to adipogenesis, browning, thermogenesis, and fatty acid oxidation, adipocyte receptors, pro- and anti-inflammatory cytokines, and components of the insulin signaling pathway. In total, 41 genes showed decreased expression due to the HFD. Interestingly, the treatment of the animals with omega-3 polyunsaturated fatty acids (FO) prevented the extensive list of genes negatively affected by the HFD. Moreover, a total of 32 genes were up-regulated by the FO treatment, bringing the data from these animals into closer alignment with that of the CO group and distinctly segregating them from the group of obese mice. Taken together, our results clearly indicate a significant impact of FO treatment on the chemokine and cytokine signaling-mediated inflammation pathway in WAT extracted from mice receiving the HFD.

Furthermore, among the genes altered in the array, we were particularly intrigued by the reversal of leptin expression following FO treatment. Leptin, a cytokine crucial for regulating energy expenditure in adipocytes, is significantly elevated in individuals with obesity and chronic inflammation. In this study, leptin expression was up-regulated by 25-fold in the HFD group compared to the control group. Remarkably, FO treatment not only reversed this increase but also reduced leptin expression in the visceral Epi WAT to values below those of the control group.

Leptin was shown to reduce ACL enzyme expression in hepatocytes and adipocytes [35]. Likewise, animal models of HFD-induced obesity typically exhibit reduced ACL expression in both hepatocytes and adipocytes [34,35,36], supporting our findings. Physiologically, the abundance of dietary lipids (from HFD) leads to decreased demand for de novo lipogenesis and, consequently, reduced ACL expression/activity, while the esterification of fatty acids to form triglycerides may remain high due to the excess of exogenous fatty acids from the diet.

Acetyl-CoA, the product of ACL, is also a substrate for histone acetyltransferases [33], including CREBBP and EP300, which promote H3K27 acetylation. This connection links its metabolism to protein acetylation processes. Thus, the decreased expression of Acly and ACL may imply a reduction in cellular acetyl-CoA concentration, limiting substrate availability for histone acetylation. This phenomenon is likely to have occurred in our model and is consistent with another study [34]. Furthermore, it has been demonstrated that ACL plays a role in regulating gene expression by influencing histone acetylation [31,38,40].

Herein, simultaneous with the observed exacerbation in leptin expression, we noted a synchronized down-regulation of Acly/ACL alongside a decrease in global H3K27ac levels in the visceral WAT of obese animals. Taken together, our findings suggest a plausible correlation between the reduction in ACL (induced by HFD) and the subsequent decrease in substrate availability for acetylase enzymes Ep300 and Crebbp, whose mRNA were likely up-regulated by a regulatory feedback system, ultimately leading to the observed decrease in H3K27ac. This reduction could potentially impact the wide range of genes negatively affected by the HFD, given the crucial role of H3K27ac in transcriptional activation and chromatin accessibility.

Significantly, the substantial up-regulation of leptin expression, as well as the down-regulation observed in ACL and H3K27ac, were effectively reversed by FO treatment. Hence, this reversal in H3K27ac may contribute to reinstating the regulation of genes commonly silenced in obesity, as evidenced in previous studies [31] and in our present array analysis. FO treatment not only prevented the extensive list of genes negatively affected by the HFD (a total of 41 genes) but also up-regulated 32 genes, aligning the expression profile of these animals more closely with that of the control (CO) group and distinctly segregating them from the group of obese mice. It is worth emphasizing that FO completely reversed the decline in H3K27ac levels, underscoring its potential to mitigate the epigenetic changes associated with obesity and suggesting a protective effect against the detrimental impacts of HFD-induced obesity on gene expression profiles. If the consumption of an HFD affects histone acetylation levels in WAT, and if this effect is mitigated by FO treatment, then gene expression programs related to inflammation and adipogenesis could potentially be influenced, as suggested by our findings.

In line with our findings, diets rich in n-3 PUFAs (EPA and DHA) have shown beneficial effects in cancer cell lines and patients, mainly by reducing inflammation through epigenetic mechanisms (for a recent review, see [41,42]). This results in global hyperacetylation of histones at N-terminal regions and specific loci [43]. The n-3 PUFA-rich diet also inhibits the enzyme ACC (Acetyl-CoA Carboxylase), increasing free acetyl-CoA, thus providing more substrate for histone acetylation [34,44]. Additionally, these diets modify the expression levels of various microRNAs, likely through changes in chromatin accessibility via histone hyperacetylation [37].

However, the specific mechanisms by which n-3 PUFA-rich fish oil regulates epigenetic marks in WAT, adipocytes, or ASCs remain unknown. The analyses reported herein were conducted in WAT, making it challenging to dissociate the specific contributions of adipocytes and ASCs. “Unhealthy” obesity is linked to significant alterations in WAT, whereas under “healthy” conditions, ASCs undergo adipogenesis and differentiate into mature adipocytes to maintain renewal [45]. Evidence suggests that WAT homeostasis is disrupted in an obesogenic context due to dysregulated adipogenesis [46], with ASCs playing a crucial role in WAT remodeling during obesity [47].

Although little is known about the molecular mechanisms behind obesity, it is understood that epigenetic regulation plays a significant role before the onset of the disease. Studies provide evidence that epigenetic dysfunction of ASCs is a key and potential regulatory event in obesity, leading to impaired adipocyte maturation and reduced adipogenic potential [48,49]. Moreover, WAT from obesity and/or type 2 diabetes individuals contains a dysfunctional pool of ASCs [50,51,52,53]. According to recent work [48], DNA methylation patterns are essentially preserved during cell commitment to adipogenesis, but obesity preconditions ASCs with a dynamic alteration of DNA methylation in selected regions, leading to WAT dysfunction and the development of metabolic syndromes in obesity. These epigenetic changes are largely influenced by environmental factors predisposing to obesity in the ASC niche. While these studies have highlighted the crucial role of ASCs in the process, epigenetic changes in response to potential modulating agents in ASCs, under physiological or pathological conditions, remain largely unexplored.

Our findings in the whole WAT encouraged us to further explore the ASCs niche. ASCs were isolated from CO and HFD groups of animals. Prior studies have underscored the influence of leptin on ASCs’ proliferation and differentiation [29,54,55]. In WAT from HFD obese animals, ASCs reside in an environment chronically enriched with leptin, believed to exert effects through paracrine signaling by binding to its receptors. We identified that ASCs express both Lepr1 and Lepr2 isoforms of the leptin receptor. Interestingly, in vitro leptin treatment did not alter the expression of Ob-Rb, the long isoform of the leptin receptor, in ASCs from obese mice, suggesting stable receptor expression despite the conditions of obesity.

We next assessed the influence of leptin on Acly expression in ASCs. We have recently demonstrated that Acly is highly expressed in ASCs and, similar to our findings in WAT, is negatively regulated by an HFD [56]. There is one study demonstrating that ACL links cellular metabolism to histone acetylation during 3T3-L1 adipocyte differentiation, and that Acly silencing impairs histone acetylation and expression of select genes [31]. Interestingly, we observed a significant decrease in Acly expression in ASCs cultivated in the presence of leptin. This finding suggests that chronic exposure of ASCs to a leptin-rich environment, as observed in obesity WAT, may contribute to the down-regulation of Acly, as detected in our study. These findings underscore an emerging role for leptin signaling in modulating gene expression patterns in ASCs and unveils a plausible mechanism that influences WAT dysfunction in obesity. Further investigation into the specific mechanisms underlying H3K27 acetylation and H3K27 methylation dissociating ASCs from adipocytes and macrophages may provide valuable insights into the pathophysiology of obesity-related dysfunction in WAT. We are now extending this research in an ongoing study.

In summary, over a 16-week study designed to induce obesity in mice through an HFD, the diet up-regulated genes associated with inflammation and macrophage markers while suppressing genes related to adipokines, lipid biosynthesis, adipogenesis, and thermogenesis. Moreover, epigenetic analysis showed changes in the expression of H3K27 enzyme modifiers and total H3K27 acetylation and methylation. FO supplementation during the last 8 weeks partially counteracted these effects, suggesting protective effects against HFD-induced obesity. FO also prevented reductions in ACL expression in obese mice, corroborating a role in preserving histone modifications. 

Overall, FO supplementation showed promise in mitigating HFD-induced obesity and influencing epigenetic and molecular pathways. Further research is needed to understand how FO affects H3K27 epigenetic marks. These collective findings emphasize the multifaceted impact of FO supplementation on metabolic and epigenetic mechanisms within adipose tissue, highlighting its potential therapeutic efficacy in ameliorating obesity-related dysregulation.

## 4. Materials and Methods 

### 4.1. Animals, Fish Oil Supplementation, and Experimental Procedure 

Eight-week-old male C57BL/6 mice were obtained from the Center for Development of Experimental Models (CEDEME), Federal University of São Paulo (UNIFESP). They were housed in a controlled environment with a 12 h light-dark cycle and a temperature maintained at 24 ± 1 °C. The experimental protocol lasted for 16 weeks, where in the first 8 weeks mice were divided into two groups: 6 control (CO) mice (9% fat, 76% carbohydrates, and 15% proteins) and 12 HFD mice (26% carbohydrates, 59% fat, and 15% proteins). After the first 8 weeks, the HFD group was further subdivided into two groups: HFD and HFD + FO (supplemented with fish oil), with 6 animals in each subgroup for the remaining 8 weeks. Fish oil supplementation, sourced from HiOmega-3 (5:1 EPA/DHA, Naturalis Nutrição and Farma Ltda, São Paulo, Brazil), was administered three times per week via oral gavage at a dosage of 2 g/kg body weight. The dosage of FO, as well as the 8-week treatment duration, were selected based on our previous studies [14,17], where the CO and HF groups also received water via gavage. Weekly monitoring of body weight and food intake was conducted throughout the 16-week period. Mice were euthanized following a 6 h fast using isoflurane anesthesia and cervical dislocation. Blood samples were obtained via orbital plexus puncture. Visceral (Epi and Rp) and subcutaneous (Ing) fat depots were excised, weighed, and then processed as detailed below.

### 4.2. Glucose Tolerance Test

Glucose tolerance tests (GTTs) were conducted in the 12th week of the study. Following a 6 h fast, mice were intraperitoneally injected with glucose (2 g kg^−1^ BW). Tail-vein glucose was measured at various intervals (15, 30, 45, 60, and 90 min) using a OneTouch^®^ glucometer. The area under the curve was calculated for each animal.

### 4.3. WAT and SVF Isolation

The Epi depot was immersed in digestion buffer and finely minced before undergoing collagenase digestion, following established protocols [57]. In brief, the minced samples were placed in a digestive buffer solution (consisting of Dulbecco’s modified Eagle’s medium—D’MEM/HEPES 20 mM/BSA 4%, and collagenase II [Sigma Chemical, St. Louis, MO, USA] at a concentration of 1.0 mg/mL, pH 7.40) and incubated for approximately 45 min at 37 °C with orbital agitation (150 rpm). Subsequently, the homogenate was filtered through nylon mesh (Corning, Oneonta, NY, USA) and centrifuged at 400× *g* for 1 min, yielding two distinct fractions: 1. supernatant, containing the isolated mature adipocytes; 2. remaining filtrate containing the stromal vascular fraction (SVF), which was further processed by centrifugation at 1500× *g* for 10 min to form a cellular pellet. This pellet was then washed twice and aspirated. Next, the SVF was incubated on ice for 10 min with a red blood cell lysis buffer (Roche Diagnostics GmbH, Mannheim, Germany) before being washed with PBS and subjected to centrifugation once again.

### 4.4. Isolation of ASCs and Leptin treatment

We followed the procedure as previously described [56]. In summary, the cellular pellet (SVF) obtained was suspended in culture medium [D’MEM Han’s F-12, supplemented with 10% fetal bovine serum (FBS) and 10 mL/L penicillin/streptomycin (Gibco BRL, Oneonta, NY, USA)] and plated in culture dishes (100 mm), which were then placed in a 5% CO_2_ incubator at 37 °C (with medium change every two days) until reaching 70–80% confluence. Subsequently, the medium was aspirated, and the plates were rinsed with PBS. The cells were detached for the first time (P1), resuspended in the same culture medium, seeded into new culture dishes for expansion, and grown until they reached 70–80% confluence again. The final step for ASC isolation involved selecting the adherent cell population within the SVF. Cell concentration was determined using a Neubauer chamber, and the cells were replated (P2). Between passages P2 and P5, cells were seeded for experiments (1 × 10^5^ cells/well in 6-well plates (35 mm)), ensuring that confluence did not exceed 80%. When cell density reached 85–90%, cells were harvested for mRNA and protein extraction either before or after treatment with leptin at a concentration of 100 ng/mL (Sigma Chemical, St. Louis, MO, USA), dissolved in culture medium, for 24 h. Each pooled cell was counted as one sample.

### 4.5. RNA Extraction and Quantitative Real-Time Polymerase Chain Reaction (qRT-PCR)

Total RNA was isolated from the entire Epi adipose depot or ASCs, utilizing Trizol reagent (Invitrogen Life Technologies, Walthamm, MA, USA). RNA quality was assessed by measuring ratios at 260/280 and 260/230 nm using NANODROP One microvolume UVVis Spectrophotometers (Thermo Scientific, Walthamm, MA, USA). Reverse transcription to cDNA was performed with the Superscript III cDNA kit (Thermo Scientific, USA). Gene expression was quantified using quantitative real-time polymerase chain reaction (PCR) with the Rotor gene (Qiagen, Dusseldorf, Germany) and SYBR Green fluorescent dye, as described in a previously published study [56,58]. Real-time PCR data analysis was conducted using the 2^∆∆Ct^ method, and the results were expressed as the ratio of target gene expression to housekeeping genes (Gapdh and 36b4). Specific primer sequences are included in Table 2.

### 4.6. PCR Array Gene Expression Analysis

For PCR array gene expression analysis, RNA was isolated using an RNA extraction kit following the manufacturer’s instructions, and the assay was conducted as previously described [59]. Briefly, cDNA and RT2 SYBR^®^ Green qPCR Mastermix (Cat. No. 330529) were utilized on a Custom Mouse RT2 Profiler PCR Array (CLAM30774R; Qiagen) consisting of 84 genes. This array allowed us to assess the expression pattern of genes encoding pro/anti-adipogenic, pro/anti-lipogenic and lipolytic, pro/anti-browning, adipokines, receptors, and components of adipocyte transduction pathways (see Table 3). CT values were exported and uploaded to the manufacturer’s data analysis web portal at http://www.qiagen.com/geneglobe. Samples were categorized into control and test groups, and CT values were normalized based on a manual selection of reference genes. Fold Change was calculated using the 2^∆∆Ct^ method via the data analysis web portal (and exported at GeneGlobe^®^, Qiagen).

### 4.7. Western Blot

Proteins from Epi WAT depots (whole tissues) or ASCs were resolved by 12% or 15% sodium dodecyl sulfate (SDS)-polyacrylamide gel electrophoresis (PAGE) and transferred to nitrocellulose membranes (0.2 μm). The membranes were blocked with 5% albumin for 2 h at room temperature and incubated with primary antibodies (anti-H3K27ac #ab4729, H3K27me3 #sab5700166, anti-ACL #ab40793, anti-Leptin Receptor #ab5593, Abcam, Waltham, MA, USA) at 4 °C overnight. Membranes were washed and incubated with secondary horseradish peroxidase-conjugated anti-rabbit IgG (Cell Signaling^®^, Danvers, MA, USA—#7074) antibodies at room temperature for 1 h. Protein blots were visualized by using an enhanced chemiluminescence Western Blotting detection kit (ECL Prime Western Blotting System, Amersham Biosciences^®^, Leicestershire, UK). Beta-actin (Cell Signaling^®^, Danvers, MA, USA—#4967L) levels were used as endogenous standard. Protein quantification was analyzed by using Scion Image software (Scion Corporation, Frederick, MD, USA). All the results were expressed relative to control group levels and corrected by the expression of the constitutive beta-actin and total protein by Ponceau.

### 4.8. Statistical Analysis

The data were analyzed using one-way Analysis of Variance (ANOVA), followed by Tukey’s post-test for comparisons between groups or Student’s *t*-test, as indicated in the figures. The results are expressed as mean ± standard error of the mean (SEM). Differences were considered significant for *p* < 0.05. Statistical analysis was performed using GraphPad Prism software version 9.1.2 (GraphPad Software Inc., San Diego, CA, USA). The PCR Array data were analyzed using the RT2 Profiler PCR Array Data Analysis Software version 3.5 (SABiosciences, Frederick, MD, USA), with tabulation and analysis in databases such as Gene Ontology Analysis and UniProt.

## 5. Conclusions

Our study highlights the significance of epigenetic modifications, specifically H3K27ac and H3K27me3, in WAT dysfunctions related to obesity. The inflammatory environment including obesity-rich leptin in obesity triggers these modifications, likely impairing adipogenesis and/or adipocyte maturation, potentially affecting gene expression patterns and exacerbating the chronic inflammatory state in a positive feedback loop. Given the limited literature on the link between FO treatment and WAT epigenetics, our findings indicate the potential role of FO in modulating these epigenetic factors and suggest a beneficial approach for addressing obesity-related complications. Future investigations will focus on exploring the reciprocal relationship between inflammation and H3K27 epigenetic marks during adipogenesis to provide further insights into how this dysregulation contributes to the persistent inflammatory state seen in obesity, as well disease pathogenesis and potential therapeutic targets.

## Figures and Tables

**Figure 1 pharmaceuticals-17-00861-f001:**
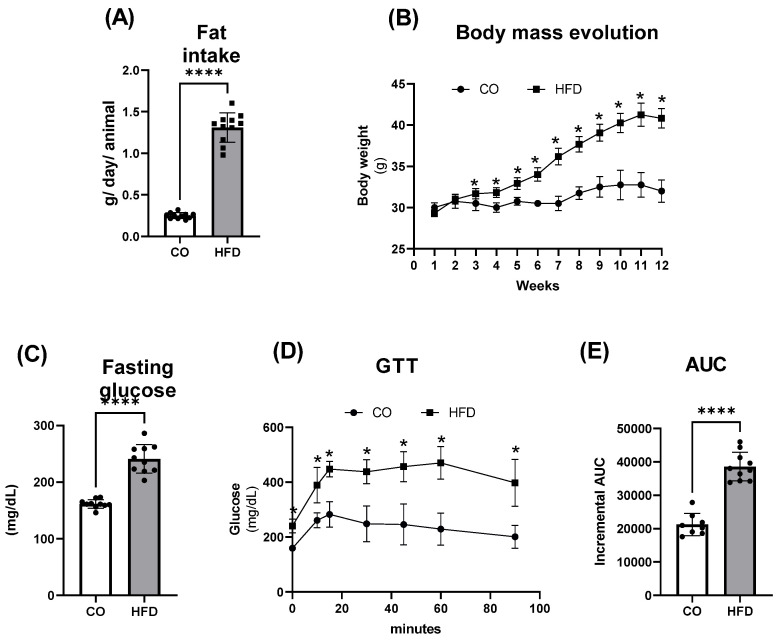
Obesity Model Characterization. (**A**) Caloric (kcal/day/animal), Food and Fat (g/day/animal) intake, (**B**) Body mass evolution, (**C**) Fasting glucose, (**D**) Glucose tolerance test or GTT, and (**E**) Incremental area under the glycemic curve in control (CO) and obese animals induced by a high fat diet (HFD) for 12 weeks. In (**A**,**B**), the measurements were performed weekly throughout the experimental protocol. In (**C**–**E**), the glycemic curve or glucose concentration versus time was calculated after glucose administration (2 g/Kg b.w.). Data were analyzed using Student’s *t*-test, and show mean ± SEM (*n* = 12). * *p* < 0.05 or **** *p* < 0.0001 versus control.

**Figure 2 pharmaceuticals-17-00861-f002:**
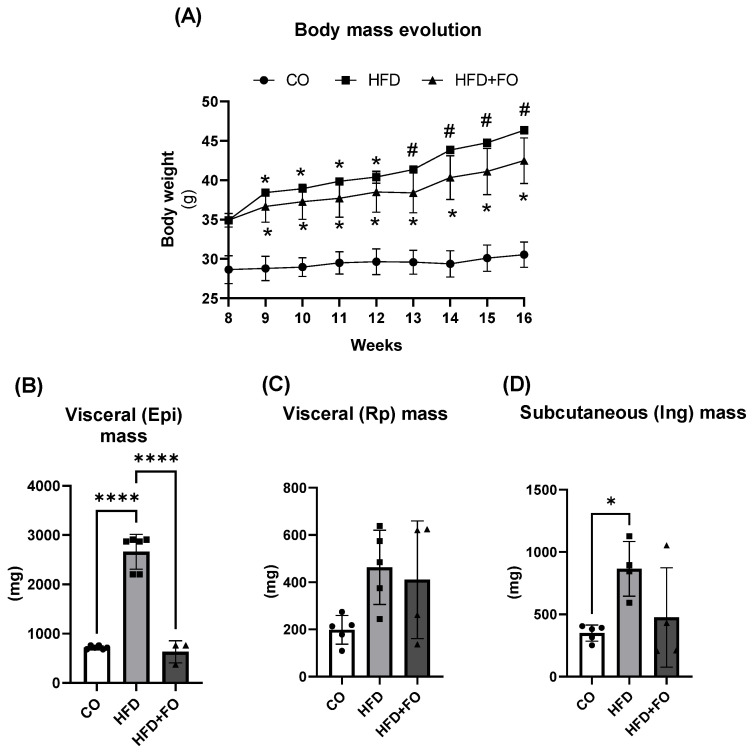
Body mass evolution (**A**), depot mass of visceral epididymal (Epi) (**B**), retroperitoneal (Rp) (**C**), and subcutaneous inguinal (Ing) (**D**) adipose tissues in milligrams (mg), after 16 weeks of experimental diets and fish oil (FO) supplementation. In the initial 8 weeks, the animals were submitted to either a control (CO) or high-fat diet (HFD). During the last 8 weeks of the experimental protocol, the diets were continued, and the animals underwent gavage (CO and HFD groups received water, while the HFD + FO group received FO) three times a week. Data were analyzed using one-way Analysis of Variance (ANOVA) followed by Tukey’s post-test, and show mean ± SEM (*n* = 6). * *p* < 0.05 vs. CO, ^#^
*p* < 0.05 vs. CO and HFD + FO, **** *p* < 0.0001.

**Figure 3 pharmaceuticals-17-00861-f003:**
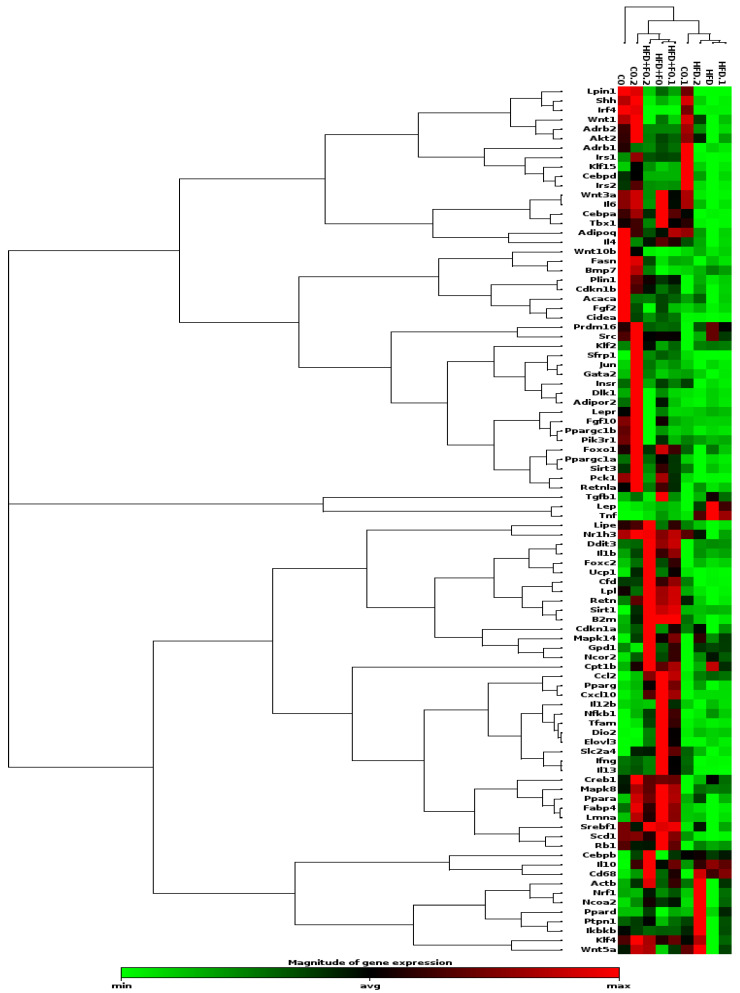
Heatmap of gene expression in Epi WAT from mice subjected to 16 weeks of experimental diets and fish oil supplementation. Control diet (CO), high-fat diet (HFD), and high-fat diet plus fish oil (HFD + FO). The gene expression level is indicated using a color scale, where red indicates higher expression and green indicates lower expression.

**Figure 4 pharmaceuticals-17-00861-f004:**
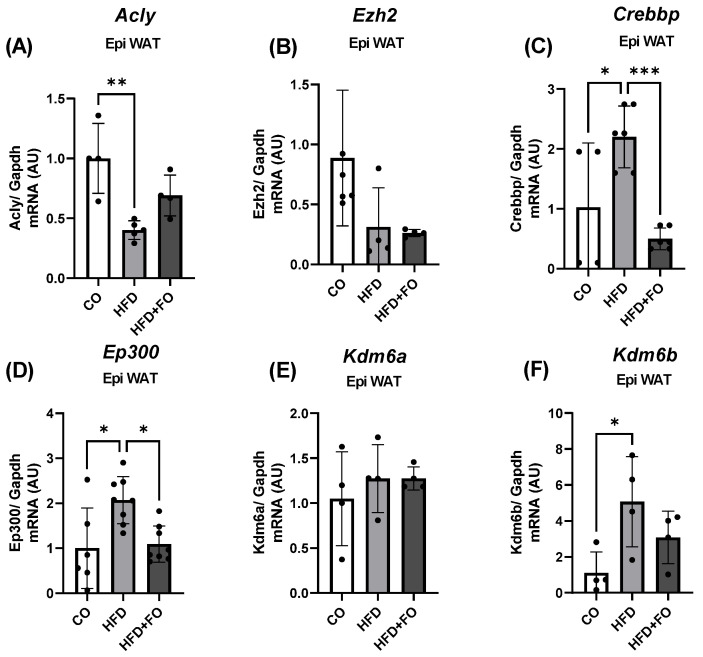
Gene expression of *Acly* (**A**) and genes encoding histone modifiers *Ezh2* (**B**), *Crebbp* (**C**), *Ep300* (**D**), *Kdm6a* (**E**), and *Kdm6b* (**F**), in the visceral Epi WAT from animals that received control diet (CO), high-fat diet (HFD), or HFD and fish oil (HFD + FO). Target genes were normalized by the constitutive *Gapdh.* Data were analyzed using one-way ANOVA followed by Tukey’s post-test, and show mean ± SEM (*n* = 4–6). * *p* < 0.05 or ** *p* < 0.01 or *** *p* < 0.001.

**Figure 5 pharmaceuticals-17-00861-f005:**
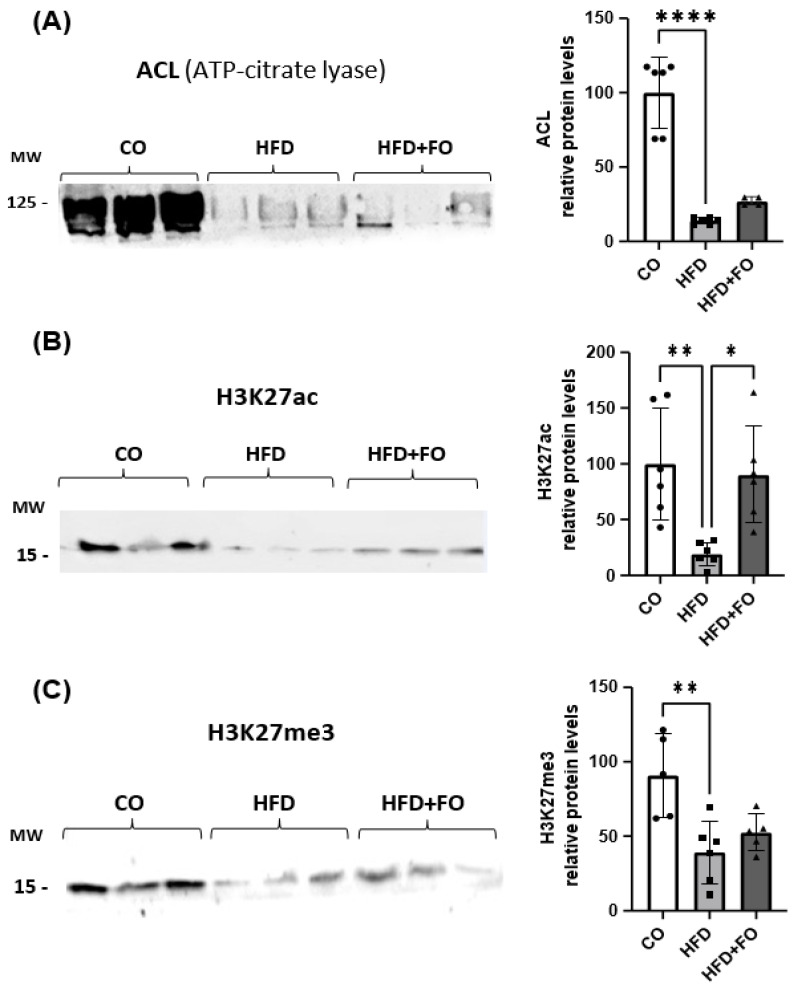
Graphical representation of the protein content of ACL (**A**), H3K27ac (**B**), and H3K27me3 (**C**), in visceral Epi WAT from animals that received a control diet (CO), a high-fat diet (HFD), or an HFD diet and fish oil (HFD + FO). Data were analyzed using one-way ANOVA followed by Tukey’s post-test. Values were expressed as mean ± SEM, in relation to the control and corrected by the expression of the constitutive beta-actin and total protein by Ponceau. A representative image of protein expression levels from 2 independent experiments is shown above each graph (*n* = 3 animals) quantified by ImageJ (Software v1.5.4e). * *p* < 0.05 or ** *p* < 0.01 or **** *p* < 0.0001.

**Figure 6 pharmaceuticals-17-00861-f006:**
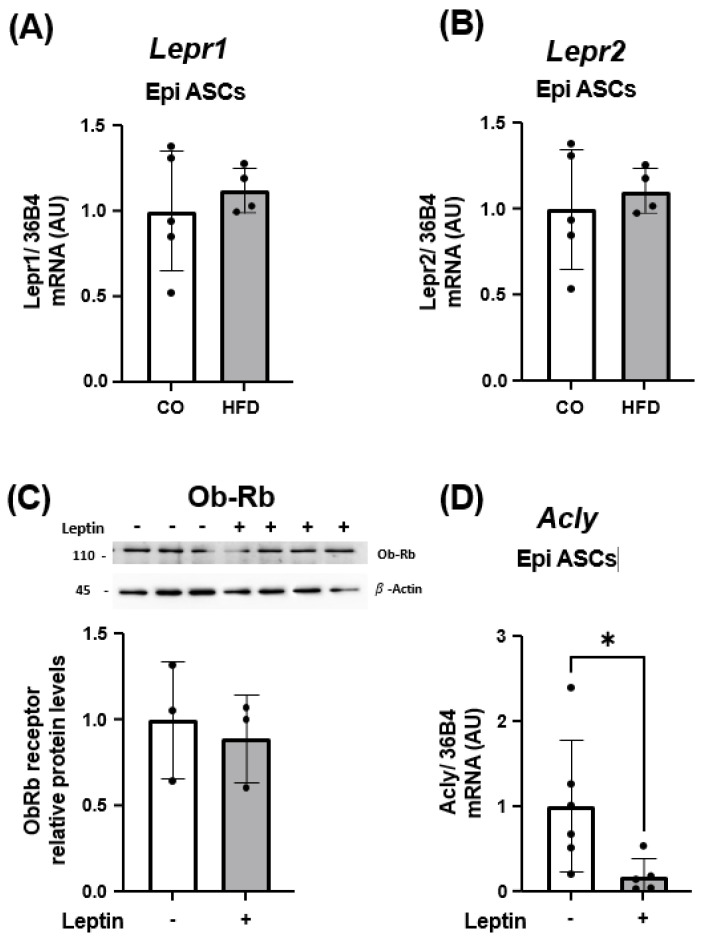
Gene expression of *Lepr1* (**A**) and *Lepr2* (**B**) in ASC isolated from WAT of animals that received a control diet (CO) or a high-fat diet (HFD). Total content of LEP R1 (Ob-Rb) protein (**C**) and gene expression of *Acly* (**D**) in ASC isolated from WAT of HFD-induced obese animals, treated in vitro with 100 ng/mL leptin for 24 h. Data were analyzed using Student’s *t*-test, and show mean ± SEM (*n* = 4–6). In (**A**–**C**), target genes were normalized by the constitutive *36B4*. In C, total content of protein was quantified by *ImageJ* and expressed in relation to the control and corrected by the expression of the constitutive beta-actin and total protein by Ponceau. A representative image of protein expression level is shown above the graphic. * *p* < 0.05.

**Table 1 pharmaceuticals-17-00861-t001:** List of genes that were up-regulated and down-regulated, comparing the obesity group to the control group: HFD vs. CO.

Gene	RefSeq Number	Fold Regulation	*p*-Value	Pathway Related
Up-regulated				
*Lep*	NM_008493	25.48	0.046332	Adipokines
*Ncor2*	NM_001253904	3.08	ns	Anti-Browning
*Dio2*	NM_010050	3.98	0.000554	Pro-Browning, fatty acid thermogenesis, and oxidation
*Elovl3*	NM_007703	2.63	ns	Pro-Browning, fatty acid thermogenesis, and oxidation
*Ccl2*	NM_011333	5.07	0.009529	Cytokines, growth factors, and signal transduction
*Il10*	NM_010548	2.08	ns	Cytokines, growth factors, and signal transduction
*Tgfb1*	NM_011577	3.06	ns	Cytokines, growth factors, and signal transduction
*Tnf*	NM_013693	10.65	0.009005	Cytokines, growth factors, and signal transduction
*Nfkb1*	NM_008689	2.52	ns	Cytokines, growth factors, and signal transduction
*Cd68*	NM_009853	9.16	0.000279	Cytokines, growth factors, and signal transduction
Down-regulated				
*Adipoq*	NM_009605	−2.82	0.010125	Adipokines
*Cfd*	NM_013459	−2.54	0.004805	Adipokines
*Retn*	NM_001204959	−3.30	0.016985	Adipokines
*Acaca*	NM_133360	−2.55	ns	Lipases and lipogenic enzymes
*Scd1*	NM_009127	−2.90	0.042774	Lipases and lipogenic enzymes
*Lpin1*	NM_001130412	−8.55	0.001238	Lipases and lipogenic enzymes
*Pck1*	NM_011044	−5.98	ns	Lipases and lipogenic enzymes
*Fasn*	NM_007988	−3.87	ns	Lipases and lipogenic enzymes
*Cebpa*	NM_007678	−2.45	0.002073	Pro-adipogenesis
*Cebpd*	NM_007679	−3.64	0.041382	Pro-adipogenesis
*Fabp4*	NM_024406	−2.01	ns	Pro-adipogenesis
*Fgf2*	NM_008006	−2.37	ns	Pro-adipogenesis
*Fgf10*	NM_008002	−2.71	ns	Pro-adipogenesis
*Jun*	NM_010591	−2.05	ns	Pro-adipogenesis
*Sfrp1*	NM_013834	−2.98	ns	Pro-adipogenesis
*Klf15*	NM_023184	−4.62	ns	Pro-adipogenesis
*Adrb2*	NM_007420	−6.37	0.001464	Anti-adipogenesis
*Dlk1*	NM_001190703	−2.36	ns	Anti-adipogenesis
*Foxo1*	NM_019739	−2.26	ns	Anti-adipogenesis
*Shh*	NM_009170	−18.41	0.000001	Anti-adipogenesis
*Wnt1*	NM_021279	−4.60	0.001147	Anti-adipogenesis
*Wnt3a*	NM_009522	−11.21	0.000001	Anti-adipogenesis
*Gata2*	NM_008090	−2.32	ns	Anti-adipogenesis
*Bmp7*	NM_007557	−2.06	ns	Pro-Browning, fatty acid thermogenesis, and oxidation
*Ppargc1a*	NR_027710	−2.40	ns	Pro-Browning, fatty acid thermogenesis, and oxidation
*Ppargc1b*	NM_133249	−2.29	ns	Pro-Browning, fatty acid thermogenesis, and oxidation
*Sirt3*	NM_001127351	−2.34	ns	Pro-Browning, fatty acid thermogenesis, and oxidation
*Tbx1*	NM_011532	−11.21	0.000001	Pro-Browning, fatty acid thermogenesis, and oxidation
*Ucp1*	NM_009463	−5.67	ns	Pro-Browning, fatty acid thermogenesis, and oxidation
*Nr1h3*	NM_001177730	−2.28	0.007073	Anti-Browning
*Wnt10b*	NM_011718	−2.50	ns	Anti-Browning
*Lepr*	NM_001122899	−2.30	ns	Adipokines receptors
*Adipor2*	NM_197985	−3.03	ns	Adipokines receptors
*Adrb1*	NM_007419	−2.73	ns	Adipokines receptors
*Ifng*	NM_008337	−11.06	0.000001	Cytokines, growth factors, and signal transduction
*Il4*	NM_021283	−2.20	ns	Cytokines, growth factors, and signal transduction
*Il6*	NM_031168	−10.93	0.000001	Cytokines, growth factors, and signal transduction
*Il13*	NM_008355	−11.21	0.000001	Cytokines, growth factors, and signal transduction
*Insr*	NM_010568	−3.52	ns	Cytokines, growth factors, and signal transduction
*Irs1*	NM_010570	−4.26	0.043055	Cytokines, growth factors, and signal transduction
*Irs2*	NM_001081212	−4.61	0.017955	Cytokines, growth factors, and signal transduction
*Pik3r1*	NM_001024955	−2.33	ns	Cytokines, growth factors, and signal transduction
*Irf4*	NM_013674	−11.92	0.000978	Cytokines, growth factors, and signal transduction

ns = Not significant. This indicates that, although there was a fold-regulation greater than 2 in the array, the ‘*p*’ value was higher than 0.5, likely due to a large variation resulting from a low number of biological replicates. Nevertheless, these genes may play an important role in the pathways activated by FO in WAT.

**Table 2 pharmaceuticals-17-00861-t002:** Sense and antisense primer sequences used for qRT-PCR.

Gene	5′ Primer (5′-3′)-Sense	3′ Primer (5′-3′)-Antisense
*Gapdh*	AAATGGTGAAGGTCGGTGTG	TGAAGGGGTCGTTGATGG
*Ep300 (p300)*	GTTGCTATGGGAAACAGTTATGC	TGTAGTTTGAGGTTGGGAAGG
*Ezh2*	CAGGATGAAGCAGACAGAAGAGG	TCGGGTTGCATCCACCACAAA
*Kdm6a*	GCTGGAACAGCTGGAAAGTC	GAGTCAACTGTTGGCCCATT
*Kdm6b*	CCTATTATGCTCCTGGGACA	TACGGCTTCCTCACTGTCGT
*Crebbp (Cbp)*	GACCGCTTTGTTTATACCTGC	TCTTATGGGTGTGGCTCTTTG
*Acly*	TCCGTCAAACAGCACTTCC	ATTTGGCTTCTTGGAGGTG
*36b4 (Rplp0)*	TAAAGACTGGAGACAAGGTG	GTGTACTCAGTCTCCAC AGA
*Lepr1*	CAGAATGACGCAGGGCTGTA	GCTCAAATGTTTCAGGCTTTTGG
*Lepr2*	ATTAATGGTTTCACCAAAGATGCT	AAGATCTGTAAGTACTGTGGCAT

*Gapdh*, Glyceraldehyde-3-phosphate Dehydrogenase; *Ep300 (p300)*, E1A binding protein p300; *Ezh2*, Enhancer of zest 2 polycomb repressive complex 2 subunit; *Kdm6a*, Lysine (K)-specific demethylase 6A; *Kdm6b*, Lysine (K)-specific demethylase 6B; *Crebbp*, CREB binding protein; *Acly*, ATP citrate lyase; *36b4 (Rplp0)*, ribossomal protein lateral stalk subunit P0; *Lepr1*, leptin receptor 1; *Lepr2*, Leptin receptor 2.

**Table 3 pharmaceuticals-17-00861-t003:** List of selected genes in Custom Mouse RT2 Profiler PCR Array.

Pathways	Genes
Adipokines	*Adipoq (Acrp30)*, *Cfd (Adipisin)*, *Lep (leptin)*, *Retn (Resistin)*
Lipases and lipogenic enzymes	*Acaca (Acc1)*, *Gpd1 (glycerol-3-phosphate dehydrogenase 1 (soluble)*, *Lipe (HSL)*, *Scd1 (stearoyl CoA desaturase)*, *Lpl*, *Pnpla2 (Atgl)*, *Lipin 1*, *Pck1 (phosphoenolpyruvate carboxykinase 1)*, *Fasn*
Pro-adipogenesis	*Cebpa*, *Cebpb*, *Cebpd*, *Pparg (PPAR gamma 2)*, *Srebf1*, *Fabp4 (aP2)*, *Pilin1*, *Fgf2 (bFGF)*, *Fgf10*, *Jun (c-jun ou AP1)*, *Lmna (Lamini A)*, *Sfrp1 (secreted frizzled-related protein1)*, *Slc2a4 (Glut4)*, *Klf15*, *Klf4*
Anti-adipogenesis	*Adrb2*, *Cdkn1a (p21Cip1*, *Waf1)*, *Cdkn1b (p27Kip1)*, *Ddit3 (Gadd153*, *Chop)*, *Dlk1 (Pref1)*, *Foxo1*, *Ncor2*, *Shh*, *Sirt1*, *Wnt1*, *Wnt3a*, *Gata2*, *Klf*
Pro-Browning, fatty acid thermogenesis, and oxidation	*Bmp7*, *Cidea*, *Cpt1b*, *Creb1*, *Dio2*, *Elovl3*, *Foxc2*, *Mapk14 (p38alpha)*, *Nrf1*, *Ppara*, *Ppard*, *Ppargc1a (Pgc1alpha)*, *Ppargc1b (Perc*, *Pgc1beta)*, *Prdm16*, *Sirt3*, *Src*, *Tbx1*, *Tfam*, *Ucp1*, *Wnt5a*
Anti-Browning	*Ncoa2*, *Nr1h3*, *Rb1*, *Wnt10b*
Adipokines receptors	*Lepr*, *Adipor2*, *Adrb1*
Cytokines, growth factors, and signal transduction	*Ccl2 (MCP1)*, *Cxcl10*, *Ifng*, *Il1b*, *Il4*, *Il6*, *Il10*, *Il12b*, *Il13*, *Tgfb1*, *Tnf*, *Insr*, *Irs1*, *Irs2*, *Akt2*, *Ptpn1 (PTP1B)*, *Ikbkb (IKKbeta)*, *Mapk8 (JNK1)*, *Nfkb1*, *Pik3r1 (p85alpha)*, *Irf4*, *Retnla (Resistin-like alpha*, *Fizz1)*, *Cd68*

*Acaca*—Acetyl-Coenzyme A carboxylase alpha; *Actb or β-actin*—Actin, beta; *Adipoq*—Adiponectin; *Adipor2*—Adiponectin receptor 2; *Adrb1*—Adrenergic receptor, beta 1; *Adrb2*—Adrenergic receptor, beta 2; *Akt2*—Thymoma viral proto-oncogene 2; *B2m*—Beta-2 microglobulin; *Bmp7*—Bone morphogenetic protein 7; *Ccl2*—Chemokine (C-C motif) ligand 2; *Cd68*—CD68 antigen; *Cdkn1a*—Cyclin-dependent kinase inhibitor 1A (P21); *Cdnk1b*—Cyclin-dependent kinase inhibitor 1B; *Cebpa or C/EBPα*—CCAAT/enhancer binding protein (C/EBP), alpha; *Cebpb or C/EBPβ*—CCAAT/enhancer binding protein (C/EBP), beta; *Cebpd*—CCAAT/enhancer binding protein (C/EBP), delta; *Cfd*—Complement factor D (adipsin); *Cidea*—Cell death-inducing DNA fragmentation factor, alpha subunit-like effector A; *Cpt1b*—Carnitine palmitoyltransferase 1b, muscle; *Creb1*—CAMP responsive element binding protein 1; *Cxcl10*—Chemokine (C-X-C motif) ligand 10; *Ddit3*—DNA-damage inducible transcript 3; *Dio2*—Deiodinase, iodothyronine, type II; *Dlk1 (Pref-1)*—Delta-like 1 homolog (Drosophila); *Elovl3*—Elongation of very long chain fatty acids (FEN1/Elo2, SUR4/Elo3, yeast)-like 3; *Fabp4 (aP2)*—Fatty acid binding protein 4, adipocyte; *Fasn*—Fatty acid synthase; *Fgf10*—Fibroblast growth factor 10; *Fgf2*—Fibroblast growth factor 2; *Foxc2*—Forkhead box C2; *Foxo1*—Forkhead box O1; *Gapdh*—Glyceraldehyde-3-phosphate dehydrogenase; *Gata2*—GATA binding protein 2; *Gpd1*—Glycerol-3-phosphate dehydrogenase 1 (soluble); *Ifng*—Interferon gamma; *Ikbkb (IKKbeta)*—Inhibitor of kappaB kinase beta; *Il10*—Interleukin 10; *Il12b*—Interleukin 12b; *Il13*—Interleukin 13; *Il1b*—Interleukin 1 beta; *Il4*—Interleukin 4; *Il6*—Interleukin 6; *Insr*—Insulin receptor; *Irf4*—Interferon regulatory factor 4; *Irs1*—Insulin receptor substrate 1; *Irs2*—Insulin receptor substrate 2; *Jun*—Jun oncogene; *Klf15*—Kruppel-like factor 15; *Klf2*—Kruppel-like factor 2 (lung); *Klf4*—Kruppel-like factor 4 (gut); *Lep*—leptin; *Lepr*—Leptin receptor; *Lipe (HSL)*—Lipase, hormone sensitive; *Lmna*—Lamin A; *Lpin1*—Lipin 1; *Lpl*—Lipoprotein lipase; *Mapk14*—Mitogen-activated protein kinase 14; *Mapk8 (Jnk1)*—Mitogen-activated protein kinase 8; *Ncoa2*—Nuclear receptor coactivator 2; *Ncor2*—Nuclear receptor co-repressor 2; *Nfkb1*—Nuclear factor of kappa light polypeptide gene enhancer in B-cells 1, p105; *Nr1h3*—Nuclear receptor subfamily 1, group H, member 3; *Nrf1*—Nuclear respiratory factor 1; *Pck1*—Phosphoenolpyruvate carboxykinase 1, cytosolic; *Pik3r1*—Phosphatidylinositol 3-kinase, regulatory subunit, polypeptide 1 (p85 alpha); *Plin1*—Perilipin 1; *Ppara*—Peroxisome proliferator activated receptor alpha; *Ppard*—Peroxisome proliferator activator receptor delta; *Pparg or PPARγ*—Peroxisome proliferator activated receptor gamma; *Ppargc1a*—Peroxisome proliferative activated receptor, gamma, coactivator 1 alpha; *Ppargc1b*—Peroxisome proliferative activated receptor, gamma, coactivator 1 beta; *Prdm16*—PR domain containing 16; *Ptpn1*—Protein tyrosine phosphatase, non-receptor type 1; *Rb1*—Retinoblastoma 1; *Retn*—Resistin; *Retnla*—Resistin-like alpha; *RTC*—Reverse Transcription Control; *Scd1*—Stearoyl-Coenzyme A desaturase 1; *Scr*—Rous sarcoma oncogene; *Sfrp1*—Secreted frizzled-related protein 1; *Shh*—Sonic hedgehog; *Sirt1*—Sirtuin 1 (silent mating type information regulation 2, homolog) 1 (*S. cerevisiae*); *Sirt3*—Sirtuin 3 (silent mating type information regulation 2, homolog) 3 (*S. cerevisiae*); *Slc2a4 (Glut4)*—Solute carrier family 2 (facilitated glucose transporter), member 4; *Srebf1*—Sterol regulatory element binding transcription factor 1; *Tbx1*—T-box 1; *Tfam*—Transcription factor A, mitochondrial; *Tgfb1*—Transforming growth factor, beta 1; *Tnf*—Tumor necrosis factor; *Ucp1*—Uncoupling protein 1 (mitochondrial, proton carrier); *Wnt1*—Wingless-related MMTV integration site 1; *Wnt10b*—Wingless related MMTV integration site 10b; *Wnt3a*—Wingless-related MMTV integration site 3A; *Wnt5a*—Wingless-related MMTV integration site 5A.

## Data Availability

Data are contained within the article.

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
