# Peer review of "Fish Oil Supplementation Mitigates High-Fat Diet-Induced Obesity: Exploring Epigenetic Modulation and Genes Associated with Adipose Tissue Dysfunction in Mice"

_pharmaceuticals, 2024, doi:10.3390/ph17070861_

Round 1
Reviewer 1 Report (Previous Reviewer 1)
Comments and Suggestions for Authors
The authors were improved their study with new title “Fish Oil Supplementation Mitigates High-Fat Diet-Induced Obesity: Exploring Epigenetic Modulation and Genes Associated with Adipose Tissue Dysfunction in Mice” instead of the old one "Transcriptomic and Histone 3 Lysine 27 (H3K27) Modifications in Mice Adipose Tissue are Triggered by a High-Fat Diet and Alleviated by Fish Oil Treatment", I am still believe that this work has great findings which seem valuable and suitable contribution to be published in the pharmaceuticals
· Instead of using “ treated (or not) with FO” you can mention in the abstract the divided groups of mice, how many and which one is control or not
· The introduction section was improved
· It is recommended to improve the quality of the figure’s resolution, the font there is not clear
· The conclusion section should be improved more for the main findigs presenatation and your suggested future work
· All other recommendations were solved and answered according to our recommendation in the last submission
Best wishes
Author Response
Thank you very much for taking the time to review and for the suggestion to improve our manuscript. You can find the detailed and revisions/corrections responses below.

Reviewer 2 Report (New Reviewer)
Comments and Suggestions for Authors
Dear Author,
The manuscript entitled "Fish Oil Supplementation Mitigates High-Fat Diet-Induced Obesity: Exploring Epigenetic Modulation and Genes Associated with Adipose Tissue Dysfunction in Mice" represents a significant contribution to the scientific literature on obesity and dietary interventions. The study's scientific importance, methodological rigor, and clarity of presentation commend it for publication in a reputable scientific journal. I commend the authors for their exemplary work and recommend acceptance of the manuscript for publication, with minor revisions to address any remaining editorial suggestions or clarifications.
- The manuscript tackles a significant research question with relevance to public health and biomedical science. Investigating the impact of fish oil supplementation on obesity-related epigenetic modulation and gene expression in mice represents a novel and innovative approach to understanding the complex mechanisms underlying metabolic dysfunction.
- The study's originality lies in its integration of epigenetic and gene expression analyses to elucidate the molecular pathways influenced by fish oil supplementation. By bridging these two fields, the authors provide a comprehensive perspective on the potential therapeutic effects of dietary interventions in obesity management.
- The experimental design demonstrates commendable attention to detail and rigor. Adequate sample sizes, appropriate controls, and well-defined outcome measures enhance the validity and reliability of the findings. The inclusion of epigenetic profiling adds depth to the analysis and strengthens the scientific rigor of the study.
- The authors employ state-of-the-art techniques and methodologies, including next-generation sequencing and bioinformatics analysis, to characterize the molecular signatures associated with fish oil supplementation. This methodological sophistication underscores the study's scientific rigor and positions it at the forefront of obesity research.
- The manuscript is well-written and effectively organized, facilitating clear communication of the study objectives, methods, results, and conclusions. The introduction provides necessary background information to contextualize the research, while the discussion integrates the findings with existing literature, offering insightful interpretations and implications.
- Figures and tables are thoughtfully constructed and appropriately annotated, aiding in data visualization and interpretation. The clarity of presentation enhances the accessibility of the findings to a broad readership, including researchers and clinicians interested in obesity and metabolic disorders.
Author Response
Thank you very much for taking the time to review and for the suggestion to improve our manuscript. You can find the detailed and revisions/corrections responses below.

Reviewer 3 Report (New Reviewer)
Comments and Suggestions for Authors
To revise the word "obese" to "obesity" throughout the different texts of the article. Since according to new nomenclature, it is preferable to use "obesity models," "obesity mice," "individuals with obesity," "obesity animals," "environmental factors for obesity."
Author Response
Thank you very much for taking the time to review and for the suggestion to improve our manuscript. You can find the detailed and revisions/corrections responses below.

Reviewer 4 Report (New Reviewer)
Comments and Suggestions for Authors
This study investigates the impact of fish oil supplementation on obesity induced by a high-fat diet (HFD) in mice. The research explores the epigenetic changes and gene expressions associated with adipose tissue dysfunction. Results show that fish oil mitigates the effects of HFD-induced obesity, highlighting its potential in modulating inflammation and metabolic dysfunction through epigenetic mechanisms.
Some comments for improving the manuscript:
1. Introduction:
- while the introduction provides general context on obesity and the inflammation associated with it, it may be useful to expand this section to highlight more specifically why it is important to study the modulation of inflammation in the context of obesity and how this might influence the management of obesity itself.
- This reviewer suggests to clarify the link between H3K27 acetylation/methylation and inflammatory state in the adipose tissue of an obese subject.
- This reviewer suggests adding more recent references, if available, to support the claims made.
2. Material and method:
- Line 440-442: there is a repetition.
3. Discussion:
- Consider including a more explicit section on study limitations and future directions.
- Broaden the discussion on the impact of observed epigenetic modifications and how these could translate into therapeutic interventions.
- Strengthen the connection between the results obtained in mouse models and possible applications for human health.
Comments on the Quality of English LanguageThe quality of the English in the document is generally good, with a well-organized structure and appropriate scientific terminology. However, there are some places where the fluidity and clarity can be improved.
Author Response
Thank you very much for taking the time to review and for the suggestion to improve our manuscript. You can find the detailed and revisions/corrections responses below.

Reviewer 5 Report (New Reviewer)
Comments and Suggestions for Authors
Methodology:
Please mention the N number for different mice groups
Results:
Most of the figures are blurry, pixilated and needs improvements
Western blot figures:
Fig 5: Very bad and not representative (ACL) and H3is overexposed
Same with H3K27 panel B and C
Fig 6: Loading control (actin) are not equal
All figures of WB in the original images documents are not representable
Very low N number (n=3) for WB experiment
General:
Have the authors measure the plasma levels of inflammatory mediators?
Author Response
Thank you very much for taking the time to review and for the suggestion to improve our manuscript. You can find the detailed and revisions/corrections responses below.

Reviewer 6 Report (New Reviewer)
Comments and Suggestions for Authors
The article is interesting and relevant. The problem of studying the role of fish oil in the development of obesity and associated metabolic disorders is relevant due to the increased prevalence of abdominal obesity and metabolic syndrome in the world. The study was conducted experimentally. The research design is clear, the research methods are fundamental. Important and interesting genetic results were obtained.
My comments: 1. It is necessary to update the list of references. There should be more sources from recent years (2022-2024). This is not difficult, since there is enough literature on the problem.
2. Source of literature No. 4 indicates that the publication date of the article is December 2024. This is an error because the article was published in January 2024.
3. In the “Materials and Methods” section, lines 440-441 duplicate lines 438-439. Needs to be fixed.
4. In section 4.1. It is not stated how many animals were in each subgroup in the first 8 weeks of the experiment, and how many animals were in each subgroup after dividing the fish oil group in the second 8 weeks of the experiment.
5. The conclusion of the article is very ambitious and does not accurately correspond to the results obtained. The phrases "our study underscores the significance of our findings" and "presenting a promising avenue for addressing obesity-related complications within the scope of this research." are especially immodest. We need to write a more modest conclusion.
Author Response
Thank you very much for taking the time to review and for the suggestion to improve our manuscript. You can find the detailed and revisions/corrections responses below.

Round 2
Reviewer 6 Report (New Reviewer)
Comments and Suggestions for Authors Dear editor, I looked at the article again.The authors corrected the article according to my comments.
The article has improved.
I have no further comments.
Author Response
We would like to thank the reviewer for the suggestions and corrections requested and we are grateful for answering all questions.
This manuscript is a resubmission of an earlier submission. The following is a list of the peer review reports and author responses from that submission.
Round 1
Reviewer 1 Report
Comments and Suggestions for Authors
The study titled "Transcriptomic and Histone 3 Lysine 27 (H3K27) Modifications in Mice Adipose Tissue are Triggered by a High-Fat Diet and Alleviated by Fish Oil Treatment" investigates the impact of fish oil (FO) enriched with eicosapentaenoic acid (EPA) on obesity induced by a high-fat diet (HFD) in mice. The research delves into the epigenetic modifications in white adipose tissue (WAT) and the role of adipose-derived stem cells (ASCs) in this context. The findings reveal that mice subjected to a HFD exhibited increased body mass, fat accumulation, and altered gene expression associated with WAT inflammation and dysfunction. However, FO supplementation in the last 8 weeks of the 16-week study mitigated these effects, suggesting a potential protective role against HFD-induced obesity. The analysis of H3K27 modifications in histones indicated that HFD-induced changes were partially reversed by FO treatment, highlighting the influence of FO on epigenetic pathways. Furthermore, the study explored leptin signaling in ASCs, proposing a potential mechanism for ASC dysfunction in the leptin-rich environment of obese WAT.
All of these findings seem valuable and suitable contribution to be published in the pharmaceuticals Journal after justifying the mentioned following points:
· The similarity rate is too high in the method section and it should be reduced accordingly
· What specific molecular pathways and mechanisms does the study propose through which fish oil supplementation alleviates high-fat diet-induced obesity in mice? Are there identified signaling pathways or key genes that play a crucial role in this protective effect?
· The study mentions an 8-week fish oil treatment period following a 16-week high-fat diet. How was this duration determined, and is there evidence to suggest that a longer or shorter treatment period would yield different results?
· You can add more data according to the obesity and Diabetes statistics you can use the following recent works” Assessing the therapeutic potential and safety of traditional anti-obesity herbal blends in Palestine. Sci Rep 14, 1919 (2024).” And “Biomolecules 2023, 13, 1486.” Which well improve the introduction well
· According to line 93 “As expected, there was a decrease in food and caloric intake (data not shown),” what do you mean with data not shown ??
· Given the identified relationship between leptin expression and Acly gene expression, can the study elaborate on the specific role of Acly in the context of obesity and fish oil treatment? How does the reversal of Acly expression by fish oil contribute to the observed effects on cellular acetyl-CoA concentration and histone acetylation?
Overall, the research underscores the efficacy of FO supplementation in alleviating HFD-induced obesity, impacting both epigenetic and molecular pathways, and providing insights into the involvement of ASCs and leptin signaling in WAT dysfunction associated with obesity.
Best wishes
Reviewer 2 Report
Comments and Suggestions for Authors
I read and reviewed the original article entitled: "Transcriptomic and Histone 3 Lysine 27 (H3K27) Modifications in Mice Adipose Tissue are Triggered by a High-Fat Diet and Alleviated by Fish Oil Treatment". The paper presents several flaws in both results presentation and methods that should be fixed before being considered for publication.
1. Title is uninformative. A real transcriptomic was not done.
2. Introduction should state the aim of the study and should briefly report the present literature and which gap the authors want to fill. This in unclearly stated
3. Results are presented without a real sequentially, this makes the readers feel very confused.
1 - you report phenotypic differences among the HFD and CTRL models and this is ok
2 - You notice differences in fat accumulation in different anatomical sites and you just focus on epididymal fat for the PCR array. Differences in sat e vat gene expression should be considered
3. Then you find Acly differential expressed and proceed to see acetilation and metilation od H3 - why ??
4. Then you pull out leptin and treat some ASC with leptin to find reduced acly- which ASC did you take. Why not using cells from HFD and CTRL mice? why not trying to revert ASC phenotype with fish oil.
Overall the paper looks confused, some experiment seems incomplete other should be rethinked. My advice is to reject the paper by now and reconsider a novel submission after extensive revision.
Comments on the Quality of English LanguageEnglish language is correct, minor spelling errors should be checked throughout the text. Heatmap should be corrected since samples names are reported in Portuguese.
Reviewer 3 Report
Comments and Suggestions for Authors
In “Transcriptomic and Histone 3 Lysine 27 (H3K27) Modifications 2 in Mice Adipose Tissue are Triggered by a High-Fat Diet and 3 Alleviated by Fish Oil Treatment”, the authors show that fish oil (FO) supplementation in HFD-induced obese mice leads to reduction in visceral epididymal WAT and in transcriptomic changes in epididymal WAT. Among these transcriptomic changes, alterations in the expression of epigenetic modifiers was observed as well as changes in H3K27ac.
Even though these findings are interesting, there are multiple major points that the authors need to address, to support their conclusions and provide a complete characterization of the effects in response to FO supplementation. Below are my comments regarding points that need to be addressed:
1. In Figure 1A, it is mentioned “decrease in food and caloric intake (data not shown)”. Data should be shown, there is no reason not to show them. They can be added in the supplement, if there are space limitations in the main figures.
2. In paragraph 2.2, it is mentioned “As expected [5,6], mice consuming HFD presented a significant increase of 3x in body mass when compared to the CO group, and the HFD+FO group was still 2x higher in relation to CO group, but presented a significant reduction of ~20% (P < 0.05) compared to the HFD group (data not shown)”. Why are these results not shown? These data should be shown. They can be added either in the main or supplementary figures, as it is important to show the effects of the treatment on body weight. Authors should show body weight for all weeks of treatment, not only for the end point.
3. Were any other parameters of body composition measured during the treatment period? E.g. were fat and lean mass measured using DEXA?
4. The authors should also report the weight of other tissues, in addition to visceral and subcutaneous adipose weights. Specifically, weight of liver, interscapular brown adipose tissue, and muscle weights (e.g. weight of gastrocnemius) should also be measured and reported.
5. The titles of graphs in Figure 2 should mention “adipose”
6. The authors perform GTT and measure fasting blood glucose in Figure 1, but do not measure any aspects of glucose metabolism during treatment. The authors should measure in HFD vs. HFD+FO treatment: fasting blood glucose, fasting insulin levels, fasting ketone levels, fasting free fatty acids and triglycerides. They should also perform a glucose tolerance test and an insulin tolerance test, to complete the metabolic characterization of the FO treatment.
7. The labels of the groups on the heatmap in Figure 3 should be labeled more clearly and samples should be in order by group (e.g. first controls, then HFD, then HFD+FO).
8. In Figures 1, 2, 4, 5, and 6, the individual dots of the samples should be added on the graphs.
9. It is unclear whether the observed transcriptomic changes are linked to the H3K27ac changes. The authors should either perform ChIP-Seq of H3K27ac or ChIP-PCRs for specific genes of interest.
Comments on the Quality of English LanguageSmall improvements in phrasing, nothing major